# 🎆 Hydra-SGG: Hybrid Relation Assignment for One-stage Scene Graph Generation

**Minghan Chen**[†]   **Guikun Chen**[‡]   **Wenguan Wang**[‡]   **Yi Yang**[‡*]

[†]ReLER Lab, AAII, University of Technology Sydney
[‡]ReLER Lab, CCAI, Zhejiang University

## Abstract

DETR introduces a simplified one-stage framework for scene graph generation (SGG) but faces challenges of sparse supervision and false negative samples. The former occurs because each image typically contains fewer than 10 relation annotations, while DETR-based SGG models employ over 100 relation queries. Each ground truth relation is assigned to only one query during training. The latter arises when one ground truth relation may have multiple queries with similar matching scores, leading to suboptimally matched queries being treated as negative samples. To address these, we propose Hydra-SGG, a one-stage SGG method featuring a Hybrid Relation Assignment. This approach combines a One-to-One Relation Assignment with an IoU-based One-to-Many Relation Assignment, increasing positive training samples and mitigating sparse supervision. In addition, we empirically demonstrate that removing self-attention between relation queries leads to duplicate predictions, which actually benefits the proposed One-to-Many Relation Assignment. With this insight, we introduce Hydra Branch, an auxiliary decoder without self-attention layers, to further enhance One-to-Many Relation Assignment by promoting different queries to make the same relation prediction. Hydra-SGG achieves state-of-the-art performance on multiple datasets, including VG150 (**16.0** mR@50), Open Images V6 (**50.1** weighted score), and GQA (**12.7** mR@50). Our code and pre-trained models will be released on Hydra-SGG.

## 1 Introduction

A scene graph is a data structure that describes the entities (objects) in a scene and the relations between these objects [3]. Scene graph generation (SGG) has attracted significant research attention [20, 50, 3, 43, 71] due to its ability to enhance machines' semantic comprehension of visual content. It has been widely adopted in various downstream applications, such as robotic vision and interaction [47, 60, 80, 48], image synthesis and manipulation [79, 10, 22], visual question answering [21, 66, 41], and video understanding [78, 68, 76, 81].

Mainstream SGG methods [85, 64, 50, 91, 61, 29, 35] work in a *two-stage* fashion. First, an off-the-shelf object detector extracts all entities within an image. Then, the extracted entities are permuted, yielding $N(N-1)$ entity pairs for $N$ detected entities. These entity pairs are

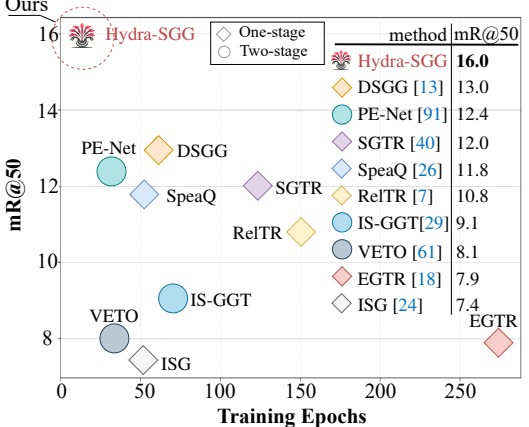

Figure 1: Comparison with other SGG methods in mR@50 and training epochs on VG150 [75].

used to predict the relationships between the corresponding entities. However, the two-stage methods face a critical limitation: they predict relations for all entity pairs, even though many pairs do

---

*Corresponding author

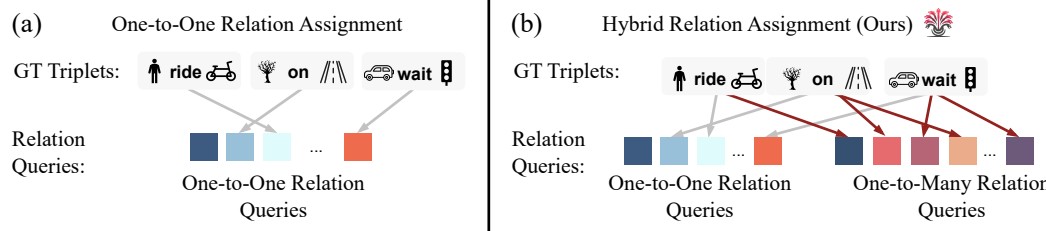

Figure 2: (a) Previous DETR-based SGG methods such as RelTR [7] and SGTR [40] match each GT relation with only one query. (b) Our Hybrid Relation Assignment utilizes both One-to-One and One-to-Many assignments, generating more positive samples and thus accelerating training.

not participate in any relations. This incurs heavy computational overhead and time-consuming inference, especially in complex scenes with numerous entities and intricate interactions.

Recently, the simplicity of DETR-based SGG methods [7, 40] has led to an ongoing paradigm shift apart from the two-stage framework. Specifically, a sequence of visual tokens, mapped from the input image, interacts with a predefined set of relation queries in a Transformer decoder for simultaneous object detection and relation prediction. Then, One-to-One set matching (*i.e.*, Hungarian matching [28]) is used to assign ground truth labels to the predictions (Fig. 2a). This set-prediction framework allows DETR-based SGG models to eliminate hand-designed components, such as Non-Maximum Suppression (NMS), which are commonly used in traditional two-stage methods.

Unfortunately, one significant drawback of DETR-based SGG models is their slow convergence. For instance, one-stage RelTR [7] requires 150 training epochs to converge, while a recent two-stage method, PE-Net [91], only needs 32 epochs. This drawback can be attributed to the sparse relation supervision induced by Hungarian matching, which assigns each ground truth to **only one** relation query. The sparsity of relation annotations per image further exacerbates this issue. For instance, VG150 [75] `train` averages only 5.5 ground truth relation triplets per image [43]. This means that each image provides a mere 5.5 positive relation queries to optimize the loss functions. In the case of RelTR [7], which has 200 predefined relation queries, only $2.75\%$ of these queries are positive samples per optimization step. Consequently, relation queries require more optimization steps to learn due to the limited number of positive samples for training. Furthermore, approximately $50\%$ [26] of the plausible but suboptimally matched queries (*i.e.*, queries with correct subject-object classification and IoU $> 0.6$ for both boxes) are simply treated as `no-relation` due to the One-to-One constraint of Hungarian matching. This constraint discards valuable supervisory signals by treating suboptimal yet informative queries as negative samples. These queries, while not the best matches for ground truth labels, may capture informative relational cues that could contribute to model learning. Simply assigning these queries as negative samples may introduce false negatives, which in turn leads to label noise and performance degradation [45].

To accelerate the training of DETR-based SGG models, we introduce Hydra-SGG[1], an efficient framework that addresses the slow convergence problem in one-stage DETR-based SGG models. The cores of Hydra-SGG are Hybrid Relation Assignment and Hydra Branch. Specifically, Hybrid Relation Assignment synergizes One-to-One and One-to-Many Relation Assignment strategies, providing over **50%** more positive queries per training step than previous arts such as RelTR [7]. To further enhance this assignment strategy, we introduce an auxiliary branch called Hydra Branch. This branch is specifically designed to encourage different queries to predict duplicated relations by removing self-attention in the decoder, creating a synergistic effect with our Hybrid Relation Assignment. The branch shares all other parameters with the original decoder, and this intentional design choice leads to improved supervision signals during training (§3.3).

Hydra-SGG makes three main contributions to the field of SGG: **First**, we propose an efficient framework that effectively addresses the slow convergence problem in one-stage DETR-based SGG models through a hybrid query-label assignment and synergistic architectural design. **Second**, our Hybrid Relation Assignment strategy significantly increases relation supervision signals by min-

---

[1]Hydra is an abbreviation for "**Hy**brid **R**elation **A**ssignment". The name is chosen because it reflects the multi-branch structure of our model, which resembles the multiple heads of the Hydra in Greek mythology.

ing false negative samples to increase positive samples during training, providing over 50% more positive samples per training step. **Third**, the proposed Hydra Branch complements our assignment strategy by encouraging duplicate relation predictions during training, while maintaining inference efficiency as it shares parameters with the original decoder and is only used during training. Hydra-SGG achieves state-of-the-art performance with remarkable training efficiency, converging $10\times$ faster than existing one-stage SGG counterparts [7, 40, 18].

Extensive experiments on three challenging SGG benchmarks, VG150 [75], Open Images V6 [30], and GQA [16], demonstrate the effectiveness of our Hydra-SGG. It achieves **16.0** mR@50 on VG150 `test` in only **12** epochs (Fig. 1), surpassing the previous state-of-the-art one-stage method SGTR [40] by **+4.0** and two-stage method PE-Net [91] by **+3.6**.

## 2 RELATED WORK

**Two-stage SGG.** Lu *et al*. [50] first proposed SGG task and designed a *two-stage* pipeline. Later, many SGG models have been built on it, using various techniques such as recurrent neural networks or its variants [85, 64, 75, 70, 58], visual translation embedding [88, 17], graph neural networks [83, 62, 74, 83, 77], external or internal knowledge integration [6, 12, 84, 1, 42, 82, 35, 4, 72], attention mechanisms [23, 53, 9], and Transformer-based models [51, 7, 40, 61, 8, 29, 59].

Despite the promising performance, two-stage SGG methods rely heavily on manually designed modules, such as NMS, anchor generation, and entity pairing generation modules [85, 64, 9]. In addition, their designs typically involve separate stages for detection, pairing, and relation classification, resulting in a complicated pipeline that cannot be trained in a fully end-to-end manner. Furthermore, these methods predict dense entity pairs in inference, leading to high time complexity and computational burden. In contrast, this paper proposes Hydra-SGG, a one-stage SGG method that offers significant advantages over two-stage methods. Hydra-SGG enables end-to-end training and surpasses the performance of two-stage methods.

**One-stage SGG.** *One-stage* methods have gained increasing attention for their simplicity and end-to-end training ability. Early works in this line adopt CNN-based one-stage detectors [46] or query-based sparse R-CNN [67] for direct relationship prediction. Recently, the DETR [2] framework significantly advances SGG and Human-Object Interaction [54, 92, 7, 40, 63, 95, 44, 25, 86, 26, 18, 38, 37, 73]. This framework enables end-to-end training by associating ground truth labels with output queries. Existing DETR-based SGG methods focus on different aspects: SGTR [40] and DSGG [13] explore query designs for relation feature extraction, SpeaQ [26] investigates relation-specific query grouping strategies to improve the specialization and discrimination of queries, while other works focus on architectural innovations [7] and lightweight frameworks [18].

However, DETR-based SGG models, while eliminating NMS, face a critical challenge of slow convergence due to sparse relation supervision, requiring significantly more training epochs than two-stage counterparts (*e.g.*, 150 for RelTR [7] vs. 32 for PE-Net [91]). Our proposed Hydra-SGG specifically addresses this limitation through a query-label assignment and architectural design while maintaining the advantages of DETR-based approaches. Hydra-SGG achieves remarkably fast convergence with only 12 epochs, surpassing both one-stage counterparts and two-stage methods to achieve state-of-the-art performance across multiple SGG datasets (16.0 mR@50 on VG150 [75], 50.1 weighted score on Open Images V6 [30], and 12.7 mR@50 on GQA [16]).

**DETR for Object Detection.** The introduction of DETR by Carion *et al*. [2] marks a significant shift away from traditional CNN-based object detection models [57, 55, 56, 45, 14], adopting an end-to-end trainable approach with a novel application of the Transformer architecture and bipartite matching. While DETR streamlines the detection process, it requires 500 epochs to converge [93]. This spurs a wave of research that focuses on improving low training efficacy, including enhancing training signal [5, 15, 19, 94], the adoption of anchor boxes [49, 52], and the leverage of efficient attention mechanism [93, 11]. For instance, Co-DETR [94] introduces extra groups of object queries and DN-DETR [31] utilizes noisy queries to increase positive samples.

In this paper, we present Hydra-SGG, a framework that addresses the sparse relation supervision challenge inherent in DETR-based SGG models. Our method introduces Hybrid Relation Assignment, which significantly increases the number of positive samples per training iteration, effectively mitigating the sparse relation supervision issue and accelerating the model's convergence.

## 3 HYDRA-SGG

§3.1 introduces our baseline model, which employs One-to-One Relation Assignment. Next, §3.2 describes Vanilla Hydra-SGG, which enhances the baseline by introducing a Hybrid Relation Assignment strategy. Finally, §3.3 details the complete Hydra-SGG model, which includes Hydra Branch (`HydraBranch`), an auxiliary decoder that promotes One-to-Many Relation Assignment. §3.4 performs a statistical analysis to validate our design of Hydra Branch, demonstrating its effectiveness in enhancing the One-to-Many assignment strategy.

### 3.1 ONE-TO-ONE RELATION ASSIGNMENT BASELINE

**Problem Formulation.** Given an input image, SGG models aim to generate a scene graph in the form of relation triplet: $\langle e_{\text{sub}}, \rho, e_{\text{obj}} \rangle$, where each entity $e_{\text{sub}}, e_{\text{obj}} \in \mathcal{E}$ is represented by a category label $c \in \mathcal{C}$ (*e.g.*, `cat`, `people`, `car`) and a bounding box $b$, and $\rho \in \mathcal{P}$ is a specific relation type (*e.g.*, `on`, `have`, `ride`). In this paper, we distinguish between the terms "relation" and "relation triplet". A relation refers to the interaction or relationship between entities, while a relation triplet represents the complete structure containing the subject $e_{\text{sub}}$, object $e_{\text{obj}}$, and their relation $\rho$.

**Baseline Architecture.** Our baseline model is composed of a `Backbone` model, an `Encoder`, and a `RelDecoder` (Relation Decoder). The relation queries interact with image tokens extracted from the input image by `Backbone` and `Encoder`. The updated queries are processed by box regression, entity classification, and relation classification heads to generate the final predictions.

- **Backbone:** `Backbone` maps an input image into a feature map that has a $H \times W \times C$ dimension, where $H$ and $W$ denote the spatial size of the feature map, while $C$ is the channel dimension (*e.g.*, 2048 or 1024).

- **Encoder:** `Encoder` captures comprehensive spatial and contextual information across the image. Before feeding the feature map into `Encoder`, a $1 \times 1$ conv layer is employed to reduce the dimension of the feature map. The enhanced image tokens are denoted as $\boldsymbol{F} \in \mathbb{R}^{HW \times 256}$.

- **RelDecoder:** In `RelDecoder`, we introduce relation queries $\boldsymbol{Q}_{\text{rel}} \in \mathbb{R}^{N \times 512}$ (■■■ ⋯ ■), which are formed by concatenating subject queries $\boldsymbol{Q}_{\text{sub}} \in \mathbb{R}^{N \times 256}$ and object queries $\boldsymbol{Q}_{\text{obj}} \in \mathbb{R}^{N \times 256}$ along the channel dimension (*i.e.*, 256). Here $N$ denotes the number of queries. `RelDecoder`$(\boldsymbol{Q}_{\text{sub}}, \boldsymbol{Q}_{\text{obj}}, \boldsymbol{F})$ works as follows:

$$\tilde{\boldsymbol{Q}}_{\text{sub}}, \tilde{\boldsymbol{Q}}_{\text{obj}} = \text{SA}([\boldsymbol{Q}_{\text{sub}}, \boldsymbol{Q}_{\text{obj}}]) \in \mathbb{R}^{2N \times 256},$$
$$\bar{\boldsymbol{Q}}_{\text{sub}} = \text{CA}(\boldsymbol{F}, \tilde{\boldsymbol{Q}}_{\text{sub}}) \in \mathbb{R}^{N \times 256}, \tag{1}$$
$$\bar{\boldsymbol{Q}}_{\text{obj}} = \text{CA}(\boldsymbol{F}, \tilde{\boldsymbol{Q}}_{\text{obj}}) \in \mathbb{R}^{N \times 256}.$$

Specifically, the subject queries $\boldsymbol{Q}_{\text{sub}}$ and object queries $\boldsymbol{Q}_{\text{obj}}$ first interact in a self-attention layer (SA) to obtain updated queries $\tilde{\boldsymbol{Q}}_{\text{sub}}$ and $\tilde{\boldsymbol{Q}}_{\text{obj}}$. Subsequently, these updated queries interact with image tokens in cross-attention layers (CA). The output subject and object queries are fed into independent box regression heads and classification heads, producing box predictions $\bar{b}_{\text{sub}}, \bar{b}_{\text{obj}} \in \mathbb{R}^{N \times 4}$ and entity class predictions $\bar{p}_{\text{sub}}, \bar{p}_{\text{obj}} \in \mathbb{R}^{N \times |\mathcal{C}|}$, respectively. The relation predictions $\bar{p}_{\text{rel}} \in \mathbb{R}^{N \times |\mathcal{P}|}$ are obtained by a relation prediction head. For example, with 300 relation queries (*i.e.*, $N = 300$) and 50 relation classes (*i.e.*, $|\mathcal{P}| = 50$), the output dimension of $\bar{p}_{\text{rel}}$ would be $300 \times 50$. Finally, the outputs of `RelDecoder` are combined to generate the predicted relation triplets $\bar{\boldsymbol{R}} = \{\langle (\bar{p}_{\text{sub}}, \bar{b}_{\text{sub}}), \bar{p}_{\text{rel}}, (\bar{p}_{\text{obj}}, \bar{b}_{\text{obj}}) \rangle_n\}_{n=1}^{N}$. Note that the $i$-th subject query is concatenated only with the $i$-th object query, and there is no permutation process.

- **One-to-One Relation Assignment Loss.** In One-to-One Relation Assignment Loss $\mathcal{L}_{\text{o2o}}$ [7, 40], Hungarian matching (HM) is employed to find the optimal correspondence between the predicted relation triplets $\bar{\boldsymbol{R}}$ and the ground truth triplets $\boldsymbol{R}$. The loss function can be formulated as:

$$\mathcal{L}_{\text{o2o}} = \mathcal{L}_{\text{HM}}(\bar{\boldsymbol{R}}, \boldsymbol{R}). \tag{2}$$

The classification losses (including entity and relation) and box regression losses are computed between matched predictions and ground truth labels, which is identical with the previous models [7, 40]. The Hungarian matching hyperparameters are the same as RelTR [7]. A detailed

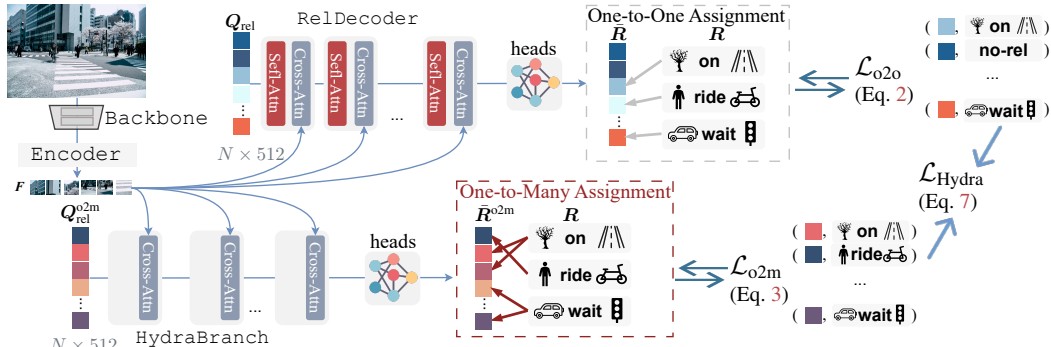

Figure 3: Overall pipeline of Hydra-SGG: For simplicity, FFN inside the Transformer layer are omitted. Hydra-SGG incorporates two Transformer decoders: `HydraBranch` and `RelDecoder`. `HydraBranch` shares its parameters with `RelDecoder` but removes self-attention layers. Hydra-SGG combines One-to-One and One-to-Many assignments in a synergy, generating more supervision signals.

implementation of the loss function, training strategy, and model architecture is provided in Appendix. The evaluation of our baseline model is given in §4.3.

## 3.2 VANILLA HYBRID RELATION ASSIGNMENT

VG150 [75] contains an average of only 5.5 ground truth relation triplets per image in `train`. Our baseline adopts a One-to-One Relation Assignment and only assigns approximately 5 out of 300 relation queries to match the ground truth relation triplets for each image (*i.e.*, $N = 300$). This severe sparse supervision reduces the training efficacy, as only about 2% of queries in each training step match with ground truth labels for learning, while the remaining 98% are treated as negative samples. Consequently, the model requires a significantly higher number of training steps to learn effectively, leading to slow convergence.

To enrich the relation supervision signals, we propose a Hybrid Relation Assignment. Specifically, we first devise a One-to-Many Relation Assignment (o2m) and then embed it into the baseline to cooperate with One-to-One Relation Assignment. Given a ground truth triplet $r = \langle (c_{\text{sub}}, b_{\text{sub}}), c_{\text{rel}}, (c_{\text{obj}}, b_{\text{obj}}) \rangle \in R$ and a predicted triplet $\bar{r} = \langle (\bar{p}_{\text{sub}}, \bar{b}_{\text{sub}}), \bar{p}_{\text{rel}}, (\bar{p}_{\text{obj}}, \bar{b}_{\text{obj}}) \rangle \in \bar{R}$, the One-to-Many Assignment score $\mathcal{S}_{\text{o2m}}$ is calculated by:

$$\mathcal{S}_{\text{o2m}}(r, \bar{r}) = \bar{p}_{\text{sub}}^{[c_{\text{sub}}]} + \bar{p}_{\text{obj}}^{[c_{\text{obj}}]} + \text{IoU}(b_{\text{sub}}, \bar{b}_{\text{sub}}) + \text{IoU}(b_{\text{obj}}, \bar{b}_{\text{obj}}). \tag{3}$$

$\mathcal{S}_{\text{o2m}}$ combines the predicted class probabilities and IoU scores for the subject and object bounding boxes. The notations $\bar{p}_{\text{sub}}^{[c_{\text{sub}}]}$ and $\bar{p}_{\text{obj}}^{[c_{\text{obj}}]}$ represent the probabilities of the ground truth classes for the subject and object, respectively. For example, if the ground truth subject class $c_{\text{sub}}$ is 3, we would extract the probability corresponding to class 3 from $\bar{p}_{\text{sub}}$.

Given $M$ ground truth relation triplets and $N$ predicted relation triplets, the final One-to-Many Relation Assignment score matrix, with dimensions $M \times N$, is computed by applying Eq. 3 to each pair of ground truth and predicted relation triplets. We keep the results with scores greater than a threshold $T$ and select the top 6 queries for each ground truth (see details in §4.3).

The proposed Vanilla Hydra-SGG applies both One-to-One and One-to-Many Relation Assignment strategies to the same set of predicted relation triplets $\bar{R}$. These triplets are obtained from the updated queries $Q_{\text{rel}}$ after passing through the model. By combining these two assignment strategies, we can formulate the vanilla version loss function as:

$$\mathcal{L}_{\text{vanilla}} = \mathcal{L}_{\text{o2o}}(\bar{R}, R) + \mathcal{L}_{\text{o2m}}(\bar{R}, R). \tag{4}$$

Our Vanilla Hydra-SGG provides richer relation supervision signals compared to the baseline by harmonizing the supervision of two assignment strategies. Specifically, it increases the number of positive samples by 65.5% in VG150 [75] `train` and 58.7% in `val` (Fig. 4a, b).Vanilla Hydra-SGG significantly improves training efficacy and performance (§4.3).

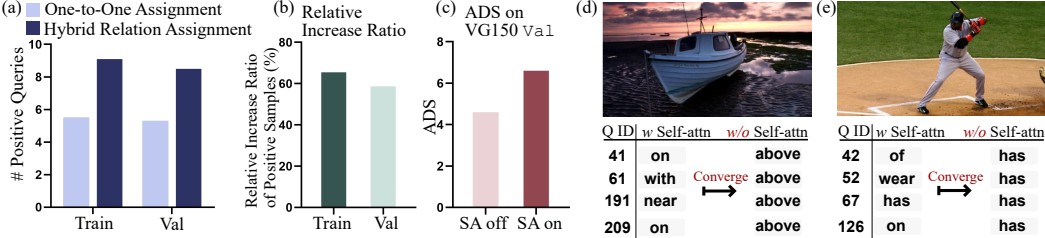

Figure 4: (a) The average number of positive samples of VG150 `train` and `val` for One-to-One and Hybrid Relation Assignment. (b) The percentage increase in positive samples achieved by Hybrid Relation Assignment compared to the One-to-One baseline. (c) ADS on VG150 `val`. (d)-(e) The visualizations show that for the same group of queries that previously predicted different relations, removing the self-attention layers causes them to make identical predictions. The Q ID column represents the ID of each relation query.

## 3.3 COMPLETE HYDRA-SGG

Each `RelDecoder` layer contains self- and cross-attention layers, and a point-wise feed-forward network (FFN). The self-attention layer enables **inter-query** interactions [69], while cross-attention and FFN do not explicitly support query interactions. Previous studies indicate that self-attention helps inhibit duplicate predictions [2, 7]. In this work, we further find that the self-attention layer in `RelDecoder` is crucial to reduce duplicated relation predictions.

We introduce a Diversity Score (DS) to quantify the impact of the self-attention layer on the diversity of relation predictions. Specifically, DS is defined as the number of distinct relation categories predicted by the model for a single image. For example, if the model predicts relations for 5 queries as (sit, sit, on, on, has), DS would be 3, as there are 3 distinct relation categories (sit, on, has). We calculate DS using the trained Vanilla Hydra-SGG model, with and without self-attention in the `RelDecoder`. Let $\text{DS}_{\text{on}}^{(k)}$ and $\text{DS}_{\text{off}}^{(k)}$ represent the DS for the $k$-th image with self-attention enabled and disabled, respectively. The average DS (ADS) across the dataset is then computed as:

$$\text{ADS}_{\text{on}} = \frac{1}{K}\sum_{k=1}^{K}\text{DS}_{\text{on}}^{(k)}, \quad \text{ADS}_{\text{off}} = \frac{1}{K}\sum_{k=1}^{K}\text{DS}_{\text{off}}^{(k)}, \tag{5}$$

where $K$ is the total number of images. A higher $\text{ADS}_{\text{on}}$ compared to $\text{ADS}_{\text{off}}$ would indicate that self-attention promotes diverse relation predictions. As shown in Fig. 4c, $\text{ADS}_{\text{off}}$ and $\text{ADS}_{\text{on}}$ of VG150 [75] `val` are 4.6 and 6.6, respectively. Fig. 4d, e illustrate how the self-attention layer enables diverse relation predictions. Without self-attention, queries converge to the same relation. These examples demonstrate self-attention's role in reducing duplicate relations.

The above findings reveal critical interactions between self-attention and One-to-Many Relation Assignment: **i) Conflict with Self-Attention:** Applying One-to-Many Relation Assignment strategy to self-attention-updated relation queries in `RelDecoder` of Vanilla Hydra-SGG causes a mismatch in optimization objectives, as self-attention promotes diversity while One-to-Many Relation Assignment assigns one ground truth to multiple queries. **ii) Benefit without Self-Attention:** Conversely, removing self-attention leads to more duplicated relation predictions, potentially synergizing with our One-to-Many Relation Assignment.

To address this potential conflict and fully harness the benefits of both the self-attention layers and the One-to-Many Relation Assignment strategy, we propose `HydraBranch` (Hydra Branch, Fig. 3), an auxiliary decoder that shares parameters with `RelDecoder` but removes the self-attention layers. This multi-branch architecture decouples the learning objectives: the self-attention layers in the main `RelDecoder` can focus on promoting diversity in relation predictions, while `HydraBranch` facilitates One-to-Many Relation Assignment. This separation enables each branch to optimize its specific function without compromising the other.

`HydraBranch` operates with the main `RelDecoder` in parallel, processing the same input but without the influence of self-attention layers. Specifically, the initial One-to-Many relation queries $Q_{\text{rel}}^{\text{o2m}}$ (■■■···■) are set to $Q_{\text{rel}}$ (*i.e.*, $Q_{\text{rel}}^{\text{o2m}} = Q_{\text{rel}}$) before being sent into `HydraBranch`. The

process of `HydraBranch`$(\boldsymbol{Q}_{\text{sub}}^{\text{o2m}}, \boldsymbol{Q}_{\text{obj}}^{\text{o2m}}, \boldsymbol{F})$ is as follows:

$$\bar{\boldsymbol{Q}}_{\text{sub}}^{\text{o2m}} = \text{CA}(\boldsymbol{F}, \boldsymbol{Q}_{\text{sub}}^{\text{o2m}}) \in \mathbb{R}^{N \times 256}, \quad \bar{\boldsymbol{Q}}_{\text{obj}}^{\text{o2m}} = \text{CA}(\boldsymbol{F}, \boldsymbol{Q}_{\text{obj}}^{\text{o2m}}) \in \mathbb{R}^{N \times 256}. \quad (6)$$

The One-to-Many predicted relation triplets $\bar{\boldsymbol{R}}^{\text{o2m}} = \{\langle(\bar{p}_{\text{sub}}^{\text{o2m}}, \bar{b}_{\text{sub}}^{\text{o2m}}), \bar{p}_{\text{rel}}^{\text{o2m}}, (\bar{p}_{\text{obj}}^{\text{o2m}}, \bar{b}_{\text{obj}}^{\text{o2m}})\rangle_n\}_{n=1}^{N}$ are derived in the same manner as $\bar{\boldsymbol{R}}$ and share prediction heads with $\bar{\boldsymbol{R}}$. Compared with Eq. 1, subject and object queries do not interact in a self-attention layer before they are sent into subsequent cross-attention layers. We apply One-to-Many Relation Assignment described in Eq. 3 to $\bar{\boldsymbol{R}}^{\text{o2m}}$ and $\boldsymbol{R}$. Hybrid Relation Assignment Loss is then given by:

$$\mathcal{L}_{\text{Hydra}} = \mathcal{L}_{\text{o2o}}(\bar{\boldsymbol{R}}, \boldsymbol{R}) + \mathcal{L}_{\text{o2m}}(\bar{\boldsymbol{R}}^{\text{o2m}}, \boldsymbol{R}). \quad (7)$$

By incorporating `HydraBranch`, the complete version of Hydra-SGG achieves **10.6** and **16.0** on mR@20 and mR@50, respectively, in just 12 training epochs, further boosting the performance compared to the vanilla version (§4.3). Note that `HydraBranch` is used only in training and discarded in inference, thus bringing no extra parameters or delay.

### 3.4 STATISTICAL ANALYSIS OF HYDRA BRANCH

To quantify how Hydra Branch enhances the One-to-Many assignment strategy, we analyze the Euclidean distances between query embedding pairs. Using 5,000 images from the VG150 `val` set, we compute pairwise distances for all 300 queries per image (totaling $\binom{300}{2} = 44,850$ pairs per image) in two conditions: with and without self-attention layers. The removal of self-attention reduces the mean query distance from 7.55 to 7.23 (5% decrease). A paired t-test confirms statistical significance ($t = 54.502$, $p < 0.001$), with a Cohen's $d$ effect size of 0.771 – approaching the threshold for a large effect ($d = 0.8$). These results demonstrate that Hydra Branch promotes query similarity through self-attention removal, thereby facilitating One-to-Many assignment.

## 4 EXPERIMENT

### 4.1 EXPERIMENTAL SETUP

**Datasets.** We conduct experiments on three datasets:

- **Visual Genome (VG150)** [27, 75] contains 150 entity and 50 relation categories. It is split into 57,723 training, 5,000 validation, and 26,446 testing images.

- **Open Images V6** [30] features 288 entity and 30 relation categories, including 126,368 training, 1,813 validation, and 5,322 testing images with relation annotations.

- **GQA** [16] encompasses 200 entity and 100 relation types, with a split of 52,623 training, 5,000 validation, and 8,209 testing images annotated for SGG tasks.

**Evaluation Metrics.** We focus on the scene graph detection (SGDet) setting on VG150 [27], GQA [16] and Open Images V6 [30] datasets. For VG150 and GQA, we report Recall@$k$ (R@$k$), mean Recall@$k$ [6, 64] (mR@$k$), and F-Recall [87] performance. mR@K calculates R@K for each predicate individually, then averages these values. It is important to note that recall metrics are more influenced by dataset bias [3], whereas mean recall provides a more holistic evaluation of the model's performance. F-Recall is the harmonic average of Recall and mean Recall. For Open Images V6, we follow evaluation protocols [30]: Recall@50, the weighted mean Average Precision (wmAP) for relationship detection (wmAP*rel*), and phrase detection (wmAP*phr*). The overall score, denoted as score$_{wtd}$, is calculated as a weighted average of these metrics: $0.2 \times \text{R@}50 + 0.4 \times \text{wmAP}_{rel} + 0.4 \times \text{wmAP}_{phr}$.

**Competitors.** Hydra-SGG is compared with methods from two categories: (1) Two-stage SGG methods, including MOTIFS [85], VCTree-TDE [65], BGNN [39], PE-Net [91], IS-GGT [29], VETO [61], UniVRD [90], DRM [32], CFA [34], and SHA [9]; (2) One-stage methods including SGTR [40], SSR-CNN [67], ISG [24], RelTR [7], DSGG [13], SpeaQ [26], and EGTR [18].

Table 1: SGDet evaluation on VG150 [27] `test` (§4.2). [+]: detector pre-trained on VG150. FPS (Frames Per Second) indicates inference speed. F-Recall of Hydra-SGG is calculated based on the best results.

| Method | Backbone | # Epoch | FPS | # Param | R@20/50/100 | mR@20/50/100 | F@20/50/100 |
|---|---|---|---|---|---|---|---|
| *Two-stage methods* | | | | | | | |
| MOTIFS [85] [CVPR2018] | ResNeXt101-FPN | - | - | 369.9M | 25.1 / 32.1 / 36.9 | 4.1 / 5.5 / 6.8 | 7.1 / 9.2 / 11.7 |
| VCTree-TDE [65] [CVPR2020] | ResNeXt101-FPN | - | - | 361.3M | 14.3 / 19.6 / - | 6.3 / 9.3 / 11.1 | 8.8 / 12.4 / - |
| BGNN [39] [CVPR2021] | ResNeXt101-FPN | - | - | 341.9M | 23.3 / 31.0 / 35.8 | 7.5 / 10.7 / 12.7 | 11.3 / 15.5 / 19.0 |
| PE-Net [91] [CVPR2023] | ResNeXt101-FPN | 32[+] | - | - | - / 30.7 / 35.2 | - / 12.4 / 14.5 | - / 17.7 / 21.2 |
| IS-GGT [29] [CVPR2023] | ResNet101 | 70 | - | - | - / - / - | - / 9.1 / 11.3 | - / - / - |
| VETO [61] [ICCV2023] | ResNeXt101-FPN | 33[+] | - | - | - / 27.5 / 31.5 | - / 8.1 / 9.5 | - / 12.5 / 14.6 |
| UniVRD [90] [ICCV2023] | CLIP ViT-B | - | - | - | - / - / - | - / 9.6 / 12.1 | - / - / - |
| DRM [32] [CVPR2024] | ResNeXt101-FPN | - | - | - | - / 34.0 / 38.9 | - / 9.0 / 11.2 | - / 14.2 / 17.4 |
| *One-stage methods* | | | | | | | |
| SGTR [40] [CVPR2022] | ResNet101 | 123[+] | - | 117.1M | - / 25.1 / 26.6 | - / 12.0 / 14.6 | - / 16.2 / 18.9 |
| SSR-CNN [67] [CVPR2022] | ResNet101 | - | - | 274.3M | 25.8 / 32.7 / 36.9 | 6.1 / 8.4 / 10.0 | 9.9 / 13.4 / 15.7 |
| ISG [24] [NeurIPS2022] | ResNet101 | 52 | - | 93.5M | - / 29.5 / 32.1 | - / 7.4 / 8.4 | - / 11.8 / 13.3 |
| RelTR [7] [TPAMI2023] | ResNet50 | 150 | 6.5 | 63.7M | 21.2 / 27.5 / 30.7 | 6.8 / 10.8 / 12.3 | 10.3 / 15.5 / 17.6 |
| DSGG [13] [CVPR2024] | - | 60 | - | - | - / 32.9 / 38.5 | - / 13.0 / 17.3 | - / 18.6 / 23.9 |
| SpeaQ [26] [CVPR2024] | ResNet101 | 52 | - | - | - / 32.9 / 36.0 | - / 11.8 / 14.1 | - / 17.4 / 20.3 |
| EGTR [18] [CVPR2024] | ResNet50 | 275[+] | 7.7 | 42.5M | 23.5 / 30.2 / 34.3 | 5.5 / 7.9 / 10.1 | 8.9 / 12.5 / 15.6 |
| *Ours* | | | | | | | |
| Hydra-SGG [ICLR2025] | ResNet50 | 12 | 5.3 | 67.6M | 21.9 / 28.6 / 33.4 $\pm 0.1$ / $\pm 0.2$ / $\pm 0.3$ | **10.3 / 15.9 / 19.4** $\pm 0.2$ / $\pm 0.2$ / $\pm 0.2$ | **14.0 / 20.5 / 24.7** |

Table 2: Evaluation on Open Images V6 [30] `test` (§4.2). [+]: detector pre-trained on Open Images V6.

| Method | Backbone | # Epoch | # Param | R@50 | $wmAP_{rel}$ | $wmAP_{phr}$ | $score_{wtd}$ |
|---|---|---|---|---|---|---|---|
| *Two-stage methods* | | | | | | | |
| Motifs [85] [CVPR2018] | ResNeXt101-FPN | - | 369.9M | 71.6 | 29.9 | 31.6 | 38.9 |
| BGNN [39] [CVPR2021] | ResNeXt101-FPN | - | 341.9M | 75.0 | 35.5 | 34.2 | 42.1 |
| PE-Net [91] [CVPR2023] | ResNeXt101-FPN | - | - | 76.5 | 35.4 | 34.9 | 44.9 |
| *One-stage methods* | | | | | | | |
| SGTR [40] [CVPR2022] | ResNet101 | 123[+] | 117.1M | 59.9 | 37.0 | 38.7 | 42.3 |
| RelTR [7] [TPAMI2023] | ResNet50 | 150 | 63.7M | 71.7 | 34.2 | 37.5 | 43.0 |
| EGTR [18] [CVPR2024] | ResNet50 | 275[+] | 42.5M | 75.0 | 42.0 | 41.9 | 48.6 |
| *Ours* | | | | | | | |
| Hydra-SGG [ICLR2025] | ResNet50 | 7 | 67.6M | $76.0_{\pm 0.2}$ | $\mathbf{42.8}_{\pm 0.2}$ | $\mathbf{44.1}_{\pm 0.2}$ | $\mathbf{50.0}_{\pm 0.2}$ |

## 4.2 QUANTITATIVE COMPARISON RESULT

**VG150** [75] `test`. Table 1 reports the comparison results on VG150 `test`. Hydra-SGG demonstrates outstanding performance on challenging SGDet, achieving mR@20 and mR@50 scores of **10.6** and **16.0**, respectively, setting a new state-of-the-art. Hydra-SGG achieves this performance with significantly shorter training time, requiring only **12** epochs. This training time is substantially shorter compared to SpeaQ [26], EGTR [18], and DSGG [13], with reductions of 40, 263, and 48 epochs, respectively. Despite the shorter training, Hydra-SGG outperforms these methods by **+4.2**, **+8.1**, and **+3.0** on the mR@50. Furthermore, it even outperforms the state-of-the-art two-stage model, PE-Net [91], by a margin of **3.6** on the mR@50 metric. We speculate that the significant improvement in mR can be attributed to our One-to-Many Assignment strategy, which ensures balanced supervision across both rare and common relations by allocating a fixed number of six queries per relation category (§ 3.2). For instance, in an image containing ten "on" relations and two "sit" relations, while One-to-One Assignment would allocate ten queries to "on" and only two to "sit", our approach assigns six queries to each, resulting in a 60% increase for common relations and a substantial 300% increase for rare relations.

**Open Images V6** [30] `test`. As shown in Table 2, Hydra-SGG achieves SOTA in only **7** epochs while SGTR [40], RelTR [7] and EGTR [18] require 119, 150, and 275 epochs. Although Open Images V6 contains more than 120,000 images, its scenes are not as complex as those in VG150 [75]. Training Hydra-SGG for 7 epochs is sufficient to achieve good performance.

**GQA** [16] `test`. Our experimental results demonstrate that Hydra-SGG achieves SOTA mean Recall performance on GQA. As shown in Table 3, Hydra-SGG achieves 12.7 and 15.9 for mR@50 and mR@100 respectively, surpassing previous best results. Our model employs the relatively lightweight ResNet50 as its backbone, in contrast to the heavier ResNeXt101 used by other methods.

Table 3: Evaluation on GQA [16] `test` (§4.2).

| Method | Backbone | R@50/100 | mR@50/100 |
|---|---|---|---|
| SHA [9] [CVPR2022] | ResNeXt101-FPN | 25.5 / 29.1 | 6.6 / 7.8 |
| VETO [61] [ICCV2023] | ResNeXt101-FPN | 26.1 / 29.0 | 7.0 / 8.1 |
| CFA [34] [ICCV2023] | ResNeXt101-FPN | - | 10.8 / 12.6 |
| *Ours* | | | |
| Hydra-SGG [ICLR2025] | ResNet50 | 23.1 / 26.8 $\pm 0.3$ / $\pm 0.3$ | **12.5 / 15.6** $\pm 0.2$ / $\pm 0.3$ |

Table 4: A set of ablative experiments about on VG150 [27] `test` (§4.3). The adopted hyperparameters are marked in red.

| Method | | mR@20 | mR@50 | # Epoch |
|---|---|---|---|---|
| Baseline | | 8.7 | 12.9 | 50 |
| Vanilla Hydra-SGG | | 9.9 (+1.2) | 14.9 (+2.0) | 12 |
| Hydra-SGG | | 10.6 (+1.9) | 16.0 (+3.1) | 12 |

(a) Key Components

| Method | | Total training time | mR@20 | mR@50 |
|---|---|---|---|---|
| RelTR [7] | | 50.0h (150 epochs) | 6.8 | 10.8 |
| Baseline | | 16.7h (50 epochs) | 8.7 | 12.9 |
| Hydra-SGG | | **12.0**h (12 epochs) | **10.6** | **16.0** |

(b) Training Time

| $T$ | | mR@20 | mR@50 | mR@100 |
|---|---|---|---|---|
| 0.3 | | 10.4 | 14.9 | 18.9 |
| 0.4 | | **10.6** | **16.0** | **19.7** |
| 0.5 | | 9.4 | 15.5 | 19.6 |
| 0.6 | | 10.3 | 14.9 | 19.1 |

(c) Threshold $T$

| # Epoch | | mR@20 | mR@50 | mR@100 |
|---|---|---|---|---|
| 10 | | 10.0 | 15.4 | 18.6 |
| 12 | | 10.6 | 16.0 | 19.7 |
| 21 | | **11.6** | **16.1** | 19.7 |
| 24 | | 10.4 | 15.2 | **19.9** |

(d) Training Epoch

| # Query | | mR@20 | mR@50 | mR@100 |
|---|---|---|---|---|
| 100 | | 8.8 | 14.1 | 17.8 |
| 200 | | 9.9 | 15.3 | 18.4 |
| 300 | | **10.6** | **16.0** | **19.7** |
| 400 | | 10.6 | 15.5 | 18.9 |

(e) Number of Relation Queries

| Loss Ratio (1-to-1 : 1-to-M) | | mR@50 | mR@100 |
|---|---|---|---|
| 1 : 0.5 | | 14.6 | 18.9 |
| 1 : 0.8 | | 15.6 | 19.4 |
| 1 : 1 | | **16.0** | **19.7** |
| 1 : 1.5 | | 15.4 | 18.7 |
| 1 : 2 | | 15.6 | 19.5 |
| 1 : 3 | | 15.4 | 19.3 |

(f) Loss Weights Between One-to-One (1-to-1) and One-to-Many (1-to-M) Losses.

| Method | | mR@50 | mR@100 |
|---|---|---|---|
| TDE [65] [CVPR2020] | | 9.2 | 11.1 |
| NICE [33] [CVPR2022] | | 10.4 | 12.7 |
| IETrans [87] [ECCV2022] | | 12.5 | 15.0 |
| CFA [34] [ICCV2023] | | 12.3 | 14.6 |
| VETO [61] [ICCV2023] | | 10.6 | 13.8 |
| NICEST [36] [TPAMI2024] | | 10.4 | 12.4 |
| Hydra-SGG [ICLR2025] | | **16.0** | **19.7** |

(g) Comparisons with Unbiasing Methods.

**Comparisons with Unbiasing Methods.** As shown in Table 4g, Hydra-SGG achieves very competitive performance on VG150 compared to these specialized debiasing methods, including TDE [65], NICE [33], IETrans [87], CFA [34], VETO [61], and NICEST [36].

## 4.3 DIAGNOSTIC EXPERIMENT

**Key Components.** We first investigate the effectiveness of our core Hybrid Relation Assignment (§3.2) and Hydra Branch (§3.3) in Table 4a. The first row gives the score of our baseline model (§3.1). The second row corresponds to the result of Vanilla Hydra-SGG, which directly applies Hybrid Relation Assignment into `RelDecoder`. The third row lists the results of the complete Hydra-SGG model. Our results show that taking the proposed Hybrid Relation Assignment in the baseline model improves the mean recall and achieves 9.9/14.9 mR@20/mR@50. Furthermore, the integration of Hydra Branch with Hybrid Relation Assignment yields a synergistic effect, boosting performance substantially. Specifically, Hydra Branch increases mR@20 from 9.9 to 10.6 and mR@50 from 14.9 to 16.0, highlighting the complementary nature of these two strategies.

**Training Efficacy.** In Table 4b, we compare the training time costs on VG150 [75]. All models are trained on eight NVIDIA RTX 4090 GPUs with a ResNet50 backbone. Our model achieves 16.0 mR@50 in 12 hours, whereas RelTR takes 50 hours to achieve 6.8 mR@50. This demonstrates the superior training efficacy of Hydra-SGG.

**Threshold $T$.** We next study the influence of the threshold $T$ in One-to-Many Relation Assignment in Table 4c. The threshold $T$ controls the quality and quantity of positive relation queries. A higher value of $T$ keeps only the high-quality positive relation queries but at the cost of reducing their number. Conversely, lowering $T$ yields more positive samples, but their quality may decrease. This trade-off between quality and quantity directly affects the final performance of the model. Experimental results show optimal performance at $T = 0.4$, yielding mR@20 of 10.6 and mR@50 of 16.0. As we increase $T$ to 0.5 and 0.6, the performance gradually decreases. This suggests that while maintaining high-quality positive samples is important, having a sufficient number of them is also crucial for the model to learn effectively.

**Training Epochs.** We further investigate the influence of training epochs on the performance. As presented in Table 4d, Hydra-SGG achieves state-of-the-art mean Recalls in just 12 epochs, demonstrating its fast convergence. We observe further improvements when increasing the epochs to 21, *i.e.*, 10.6 mR@20 and 16.0 mR@50 respectively. However, the performance decreases at epoch 24, with mR@20 and mR@50 dropping to 10.4 and 15.2, respectively. This reduction in performance can be attributed to overfitting.

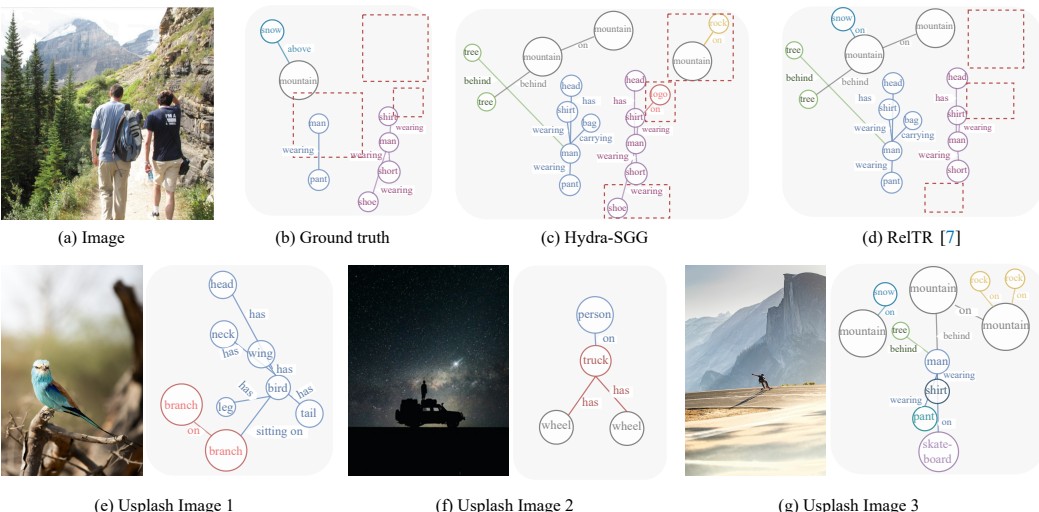

Figure 5: Qualitative results §4.4. (a)-(d) compare Hydra-SGG and RelTR [7] on a VG150 [27] `val` image. We use the same color for each entity category, and the color of a predicate matches that of its subject. Differences are highlighted with red dashed rectangles ⌐¬. (e)-(g) show scene graphs generated by Hydra-SGG from images sourced from Unsplash, a platform for freely-usable images. These images are real-world, "in the wild" scenarios, demonstrating our model's capability to handle diverse and unseen visual content.

**Number of Relation Queries.** Lastly, we studied the effect of the number of relation queries on model performance, as shown in Table 4e. The performance with 200 queries and 400 queries is inferior to that with 300 queries. The model achieved mR@20/50 of 9.9/15.3 with 200 queries, and 10.6/15.5 with 400 queries. In contrast, with 300 queries, the model reached its highest performance, with 10.6 mR@20 and 16.0 mR@50. We hypothesize that using 200 queries may be insufficient to capture the diversity of relationships within the data. On the other hand, using 400 queries could introduce excessive noise and false positives, thereby reducing the overall precision of the model.

**Impact of Loss Weights.** As shown in Table 4f, we conducted an ablation study to investigate the effect of different loss weights between One-to-One and One-to-Many losses. With a 1:1 loss ratio, the model achieves the best results (mR@50: 16.0, mR@100: 19.7). When the One-to-Many loss weight is reduced (*e.g.*, 1:0.5), the model fails to fully utilize the additional positive samples available. Conversely, increasing this weight excessively (*e.g.*, 1:3) leads to an overemphasis on One-to-Many triplets, which typically have lower quality compared to One-to-One triplets. This quality difference arises from the fundamental nature of the assignments: One-to-One assignment matches each relation label with its optimal query exclusively, while One-to-Many assignment matches a relation label with multiple queries meeting certain criteria.

### 4.4 QUALITATIVE COMPARISON RESULT

In Fig. 5a-d, we visualize the generated scene graph results on the image of VG150 [27] `val`. Both Hydra-SGG and RelTR [7] detect plausible relations, but these relations were not annotated, so these reasonable predictions become false negatives. Our Hydra-SGG detects more fine-grained relations such as `rock-on-mountain` and `logo-on-shirt`, while RelTR fails to detect such relations. In Fig. 5e-g, we visualize generated scene graph results on unseen Unsplash images.

## 5 CONCLUSION

This paper introduces Hydra-SGG, a one-stage DETR-based scene graph generation (SGG) model that addresses the slow convergence issue in existing DETR-based SGG models. The key contributions of our work are as follows: **i)** We propose a Hybrid Relation Assignment, which combines One-to-One and One-to-Many Relation Assignment strategies to increase relation supervision signals. **ii)** We introduce a Hydra Branch, an auxiliary decoder that encourages relation queries to predict duplicate relations, further enhancing the proposed One-to-Many Relation Assignment. These innovations work in synergy to significantly accelerate the learning process. Consequently, Hydra-SGG achieves state-of-the-art results on VG150 [75], GQA [16], and Open Images V6 [30] in just 12, 12, and 7 training epochs, demonstrating remarkable performance and efficiency.

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

APPENDIX

For a better understanding of the main paper, we provide additional details in this supplementary material, which is organized as follows:

- §A details the implementation details.
- §B provides the pseudo code of Hydra-SGG.
- §C shows failure cases of Hydra-SGG in VG150 [75].
- §D discusses our limitations, societal impact, and directions of future work.

## A   IMPLEMENTATION

Inspired by RelTR [7], we also set relation queries to be composed of subject and object queries. We adopt the same training techniques and attention mechanisms as previous works [31, 89, 7] in Hydra-SGG. In particular, we use anchor boxes as embeddings to accelerate training and employ deformable attention [93] respectively. We input noised box-label pairs to generate both entity and relation denoising queries. To prevent information leakage, we employ attention masks. Given the absence of an attention map in the deformable decoder, we simplify the model by omitting the convolutional mask head in RelTR [7]. In particular, we concatenate the output subject and object queries and input them into an MLP. This simplification not only reduces computational complexity but also maintains performance.

Both One-to-One and One-to-Many losses are composed of box regression loss, entity classification loss, and relation classification loss as previous works [2, 40, 7, 18, 63]. Specifically, we use Focal loss [45] for both relation and detection, as in previous works [49, 31, 89]. In the post-processing stage, the predictions are first ranked by the relation probability. Then we use the relation index to find corresponding subject and object predictions to compose the prediction triplets. Finally, we follow the same process as RelTR [7] that removes predictions where the subject and object are the same, as such data does not exist in VG150 [75], but we do not use this process in Open Images V6 [30].

For training, we adopt the same data augmentation techniques as RelTR [7] but discard random cropping since some triplets could be incomplete. The default training epochs are 12 for VG150 [75] and 7 for Open Images V6 [30]. For VG150, the learning rate is scaled by 0.1 at epoch 11, while for Open Images V6, it is scaled at epoch 6. Open Images V6 [30] contains more than 120,000 training images, but most scenes in the dataset are simpler than those in VG150 [75], therefore training for 7 epochs is enough to achieve state-of-the-art performance. Extended training on Open Images V6 may improve the performance, but considering the dataset size, the marginal benefits are limited.

Due to the inherent bias in SGG datasets, numerous unbiasing algorithms have been developed [65, 34, 33, 61, 91]. While adopting these unbiasing techniques typically leads to significant performance improvements, we opt for a fair comparison by evaluating Hydra-SGG against methods that do not employ such techniques.

## B   PSEUDO CODE

The pseudo-code of Hydra-SGG is given in Algorithm S1. Our code and pre-trained models will be made publicly available.

## C   FAILURE CASES

In this section, we present failure cases of scene graphs generated by Hydra-SGG on VG150 [75] val. In the first row (Fig. S1a-c), Hydra-SGG successfully recognizes the airplane in the image, but the predicted label is 'plane'. Although 'plane' is a plausible label, the ground truth label is 'airplane', which leads to all predicted relationships being incorrect since they don't match the ground truth. However, from a semantic perspective, the model correctly identifies these relationships and even recognizes more fine-grained relationships that are not annotated in the ground truth. For instance, the ground truth only marks the right engine of the airplane, but the model correctly identifies

---

**Algorithm 1** Hydra-SGG: PyTorch-like Pseudo-code

---

```python
# RelDecoder shares parameters with Hydra Branch
# memory: image tokens (L x D)
# Query_rel: relation queries of RelDecoder (N x D)
# Query_rel_o2m: relation queries of HydraBranch (N x D)
# Loss_o2o: apply box regression, entity, and relation classification losses between
    matched queries and GTs by Hungarian matching.
# Loss_o2m: apply box regression, entity, and relation classification losses between
    matched queries and GTs by One-to-Many Relation Assignment.

# Hydra SGG forward
def forward(sample, target):
    # extract and enhance features from the input
    memory = Encoder(Backbone(sample)) # (L x D)

    # init relation queries for RelDecoder and Hydra Branch
    Query_rel = Query_rel_o2m = self.init_queries()

    Query_rel_bar = RelDecoder(Query_rel, memory) # (N x D)
    Query_rel_o2m_bar = HydraBranch(Query_rel_o2m, memory) # (N x D)

    loss_1 = Loss_o2o(Query_rel_bar, targets)
    loss_2 = Loss_o2m(Query_rel_o2m_bar, targets)
    loss_hydra = loss_1 + loss_2
    loss_hydra.backward()

# RelDecoder
def RelDecoder(Query_rel, memory):
    Query_rel = Self_attn(Query_rel) # (N x D)
    Query_rel = Cross_attn(Query_rel, memory) # (N x D)
    Query_rel = FFN(Query_rel)
    return Query_rel

# Hydra Branch
def HydraBranch(Query_rel_o2m, memory):
    Query_rel_o2m = Cross_attn(Query_rel_o2m, memory) # (N x D)
    Query_rel_o2m = FFN(Query_rel_o2m)
    return Query_rel_o2m
```

---

Algorithm S1: Hydra-SGG core implementation.

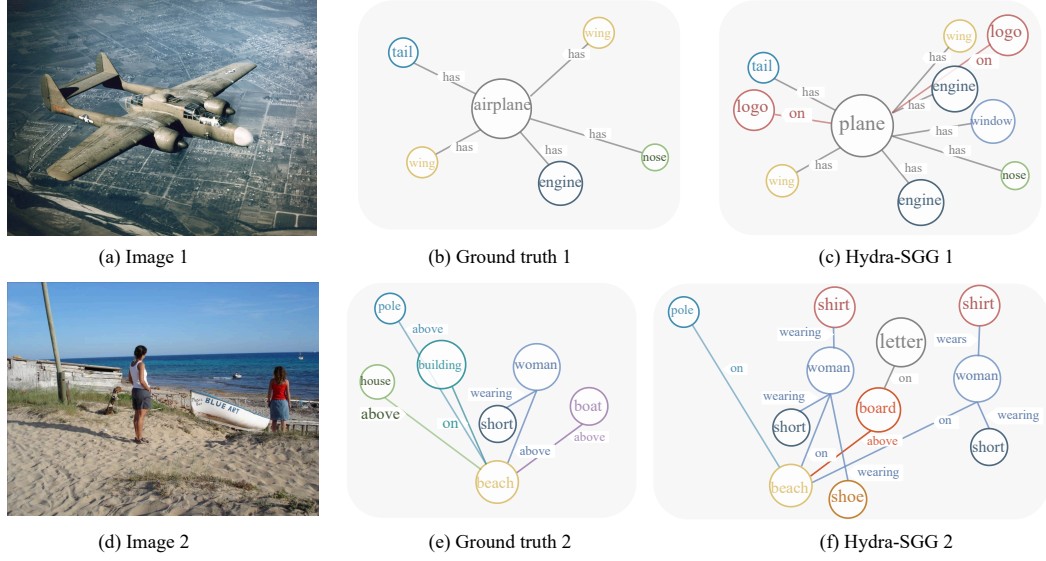

(a) Image 1     (b) Ground truth 1     (c) Hydra-SGG 1

(d) Image 2     (e) Ground truth 2     (f) Hydra-SGG 2

Figure S1: Failure cases of Hydra-SGG on VG150 [75] `val`.

both engines. In addition, the model correctly identifies the logo and windows on the airplane, which are reasonable and correct predictions but are considered incorrect because they are not annotated in the ground truth. In the second row (Fig. S1d-f), the ground truth annotation uses a single box to label two women, which is an unreasonable annotation. The model correctly predicts two women

and identifies that both women are wearing shirts and shorts, which are annotations absent in the ground truth but are semantically accurate.

# D    DISCUSSION

**Limitation Analysis.** A significant limitation of the current Hydra-SGG is its inability to predict object and relation categories beyond a predefined closed set. The algorithm can only make predictions for object and relation classes that have been explicitly defined and included in the training dataset. This constraint means that the model lacks the capability to recognize or infer novel object types or relationship categories that were not present during the training phase. Such a limitation restricts the model's applicability in real-world scenarios, where encountering previously unseen objects or relations is common. In addition, since the current One-to-Many Relation Assignment rules are manually designed, they may lack flexibility and potentially introduce some noise to the training process.

**Societal Impact.** Hydra-SGG has the potential to significantly enhance autonomous driving systems, thereby improving road safety and efficiency. For instance, when integrated into self-driving vehicles, Hydra-SGG can provide a deeper understanding of complex traffic scenarios. Beyond merely identifying vehicles, pedestrians, and road signs, it can comprehend the spatial relationships and potential interactions between these elements. This advanced scene understanding could enable autonomous vehicles to more accurately predict the behavior of other road users, leading to safer and more efficient navigation in diverse traffic conditions.

**Future Work.** A promising direction for future research is to extend the current model towards open vocabulary SGG. This advancement would allow Hydra-SGG to recognize and predict objects and relationships beyond the predefined closed set of categories used in training.

As mentioned in the limitations, since the current One-to-Many Relation Assignment rules are manually designed, they may introduce noise during training that affects learning. Therefore, another direction for future work could focus on designing better one-to-many strategies to further improve performance. For instance, we could develop a feature composition module to generate new relation triplets, or we can incorporate a relationship refine module to transfer coarse-grained relations into fine-grained relations.

