# OpenReview forum: "Hydra-SGG: Hybrid Relation Assignment for One-stage Scene Graph Generation"
_ICLR.cc/2025/Conference — ICLR 2025 Poster_

### Official Review · Reviewer_ARZw · 2024-11-01

**Soundness:** 4
**Presentation:** 3
**Contribution:** 4
**Rating:** 8
**Confidence:** 4

**Summary:**

This paper proposes a new method called Hydra-SGG for Scene Graph Generation (SGG), which addresses the challenges of sparse supervision and false negatives in one-stage SGG. To this end, it employs a one-to-many assignment strategy and a transformer decoder without self-attention called a hydra-branch. Through comprehensive experiments across various datasets and detailed ablation studies, the paper confirms the efficiency and effectiveness of this method.

**Strengths:**

1. This paper pinpoints the critical problems in one-stage SGG, which are sparse supervision and false negatives. By addressing this problem, the proposed method is superior on efficiency and effectiveness.
2. The author conducts extensive experiments, including effectiveness on diverse datasets and ablation studies, thereby validating the superiority of the proposed method. Especially, it significantly reduces the training epoch compared to the baselines.
3. The proposed method is simple yet effective.

**Weaknesses:**

The paper lacks discussion on its limitations and future works.

There are no critical points for rejection.

**Questions:**

1. While the One-to-Many alignment approach helps mitigate sparse supervision, there is a concern that it may introduce significant noise, potentially leading to an increase in false positives. What are some strategies to manage this noise effectively?
2. How would performances be impacted if the weights for One-to-One and One-to-Many alignment losses were changed? For example, I'm curious how performance would change if the weight of One-to-Many alignment loss were increased or decreased.

---

> ### Author Response · Authors · 2024-11-23
>
> We sincerely appreciate reviewer ARZw's thorough review and highly positive assessment of our work. Below, we address each point specifically.
>
> Q1: More discussion on limitations and future works.
>
> A1: We appreciate this valuable suggestion and will expand our discussion on limitations and future works in Appendix Sec D in the revised version. The new additions will be marked in italics:
>
> Limitation Analysis:
>
> > …, where encountering previously unseen objects or relations is common. *In addition, since the current One-to-Many Relation Assignment rules are manually designed, they may lack flexibility and potentially introduce some noise to the training process.*
> >
>
> Future Work:
>
> > … This advancement would allow Hydra-SGG to recognize and predict objects
> and relationships beyond the predefined closed set of categories used in training.
> >
> >
> >   *As mentioned in the limitations, since the current One-to-Many Relation Assignment rules are manually designed, they may introduce noise during training that affects learning. Therefore, another direction for future work could focus on designing better one-to-many strategies to further improve performance. For instance, we could develop a feature composition module to generate new relation triplets, or we can incorporate a relationship refine module to transfer coarse-grained relations into fine-grained relations.*
> >
>
>
>
>
> Q2:Strategies to manage noises introduced by One-to-Many Relation Assignment?
>
> A2: Thank you for this important question. Our One-to-Many Relation Assignment currently relies on heuristic rules (i.e., IoU-based), which may introduce noisy triplets. We propose two potential directions for future improvement:
>
> 1. Generate new triplets by feature composition :
>
>     Feature composition methods [ref1] can create new relation triplets at a feature level, which can be integrated into our One-to-Many Relation Assignment.
>
> 2. Refine the triplets:
>
>     We can develop a relation transfer mechanism [ref2] that focuses on refining general relations into more specific ones (e.g., "people-on-bicycle" → "people-riding-bicycle") and incorporate fine-grained triplets in One-to-Many Relation Assignment.
>
>
>
>
> Q3: How do different loss weights between One-to-One and One-to-Many affect performance?
>
> A3:  We sincerely appreciate your important suggestion. We conduct an ablation study on different loss weights between One-to-One and One-to-Many assignments. As shown in the table, setting the loss ratio to 1:1 achieves the best performance (mR@50: 16.0, mR@100: 19.7). When One-to-Many loss weight is too small (e.g., 1:0.5), the model underutilizes additional positive samples. Conversely, when the weight is too large (e.g., 1:3), it may overemphasize One-to-Many triplets, which have a relatively lower quality compared to the One-to-One triplets. In One-to-One assignment, each relation label is only matched with its optimal query, while in One-to-Many assignment, a relation label is matched with multiple queries that satisfy certain criteria. As a result, the overall quality of One-to-Many triplets tends to be lower than One-to-One triplets, since the latter only contains the most optimal matches. We add analysis in Table 4 in the revised manuscript.
>
> | Loss Ratio (One-to-One : One-to-Many) | mR@50 | mR@100 |
> | --- | --- | --- |
> | 1 : 0.5 | 14.6 | 18.9 |
> | 1 : 0.8 | 15.6 | 19.4 |
> | 1 : 1 (default) | **16.0** | **19.7** |
> | 1 : 1.5 | 15.4 | 18.7 |
> | 1 : 2 | 15.6 | 19.5 |
> | 1 : 3 | 15.4 | 19.3 |
>
> **Ref:**
>
> [ref1] Compositional Feature Augmentation for Unbiased Scene Graph Generation. ICCV 2023.
>
> [ref2] Fine-Grained Scene Graph Generation with Data Transfer. ECCV 2022.

---

> ### Comment · Reviewer_ARZw · 2024-11-25
>
> Thank you for your sincere response. All my questions have been resolved, and I also reviewed feedback from other reviewers that I had overlooked. However, I think you have addressed it well, so I will keep my score.

---

> ### Author Response · Authors · 2024-11-25
>
> Dear reviewer ARZw,
>
> We appreciate again your thoughtful review and help us further improve the quality of our work! Please let us know if any other concerns.
>
> Best regards.
>
> Authors.

---

### Official Review · Reviewer_KZeA · 2024-11-03

**Soundness:** 3
**Presentation:** 3
**Contribution:** 2
**Rating:** 6
**Confidence:** 2

**Summary:**

This paper addresses key challenges in the field of one-stage Scene Graph Generation (SGG), particularly focusing on sparse supervision and false negatives samples.To tackle these challenges, the authors propose two main contributions:
(1) Hybrid Relation Assignment: This method combines One-to-One and One-to-Many Relation Assignment strategies, increasing the number of positive samples during training and thus enhancing supervision.
(2) Hydra Branch: An auxiliary decoder branch without self-attention layers, designed to further support the One-to-Many Relation Assignment by promoting consistent relation predictions across queries.
Above methods jointly improve the training efficacy and performance of DETR-based SGG models, leading to state-of-the-art results on multiple benchmarks with significantly fewer training epochs. Overall, this paper’s motivation is clear, and the proposed method iseffective.

**Strengths:**

(1) Innovative Method Design: The proposed Hybrid Relation Assignment effectively combines One-to-One and One-to-Many matching strategies. This design helps mitigate the sparse supervision issue in existing methods and improves model convergence speed and learning efficiency.
(2) Efficient Auxiliary Branch: The Hydra Branch supports One-to-Many assignment by promoting consistent relation predictions across queries, all without adding inference-time computational costs.
(3) Good Experimental Results: The experimental results show that Hydra-SGG achieves state-of-the-art performance across multiple datasets (VG150, Open Images V6, and GQA), reaching these results in significantly fewer training epochs.

**Weaknesses:**

(1) Lacking theory/analysis, technical solutions appear to be intuitive and empirical. Faster convergence may come from more diverse samples or more training data within an epoch, because the proposed method is likely to give 16 images, but will generate more relation training pairs for training.

(2) Limited Exploration. Why removing Self-Attention benefits the One-to-Many Relation Assignment. The paper introduces the Hydra Branch without self-attention to support the One-to-Many relation assignment. However, the explanation of why removing self-attention aids this assignment is somewhat surface-level, lacking an in-depth analysis of the interaction between self-attention and One-to-Many assignment.

**Questions:**

Does faster convergence come from more diverse samples or more training data within an epoch, because the proposed method is likely to give 16 images, but will generate more relation training pairs for training?

It would be better to compare the methods with unbiased SGG methods or adopted unbiased strategies, considering the higher performance of mean Recall.

---

> ### Author Response · Authors · 2024-11-23
>
> We thank reviewer KZeA for the valuable time and thoughtful feedback. We provide a point-to-point response below.
>
> Q1: Lacking theory/analysis, technical solutions appear to be intuitive and empirical.
>
> A1: Respectfully disagree. Our work begins by identifying and analyzing a critical issue in DETR-based SGG models: the slow convergence caused by the One-to-One constraint, which results in sparse relation supervision, providing only 5.5 positive samples (L 077) per image on average for model training. This bottleneck significantly limits the model’s ability to learn effectively, highlighting a key challenge in current frameworks.
>
> **Based on this analysis, we developed our solution by drawing inspiration from well-established principles** in the community, where increasing positive training samples and reducing false negatives are widely used and shown to be effective [ref1,  ref2,  ref3]. We are the first to introduce such concepts to one-stage DETR-based SGG and explore several variations (L 066) from scratch. While our work is not purely theory-oriented, it follows a systematic approach: from problem identification, to solution design guided by established principles and empirical validation. The effectiveness of our approach is demonstrated by achieving SOTA results across multiple SGG benchmarks with only 12 epochs of training. Given such fresh insights and the substantial improvements demonstrated, this study deserves to be shared with our community.
>
> Q2: Limited exploration of how HydraBranch enhances the proposed One-to-Many assignment.
>
> A2: We first clarify the analyses and explorations that have been done on how HydraBranch enhances the proposed One-to-Many assignment.
>
> **Quantitative analysis on the optimized training signals** (L 270), We analyze the average number of positive samples in VG150 train and val for both One-to-One and Hybrid Relation Assignment, as illustrated in Fig. 4 (a). In addition, Fig. 4 (b) shows the percentage increase in positive samples achieved by the Hybrid Relation Assignment compared to the One-to-One baseline (Sec. 3.3).
>
> **Quantitative analysis on removing self-attention layers** (L 270, L 285). We introduce a Diversity Score (DS) (Sec. 3.3) to quantify the presence and absence of the self-attention layer on the diversity of relation predictions. As shown in Fig. 4 (c), our experiments demonstrated that the presence of the self-attention layer significantly increases the diversity of predicted relations, as evidenced by higher ADS (Average Diversity Score) values on VG150.
>
> **Qualitative analysis on removing self-attention layers** (L 270). We included visual examples (Fig.4 (d) (e)) illustrating how the self-attention layer affects relation predictions. These examples clearly demonstrate that without self-attention, queries tend to converge on the same relation.
>
> The above analyses showcase our efforts in designing HydraBranch and validating Hybrid Relation Assignment through both quantitative and qualitative perspectives. Our One-to-Many assignment strategy is designed to assign a single GT relation to multiple queries. To enhance this strategy, we intentionally designed a module to further encourage queries to predict duplicated relations. Through empirical analysis (L 285-L 349), we found that removing self-attention in the decoder effectively serves this purpose by making queries more similar to each other, thus creating a natural synergy with our One-to-Many assignment approach.
>
> However, we acknowledge that a deeper investigation would be valuable. To better explore how HydraBranch enhances the proposed One-to-Many assignment, we conducted additional statistical analysis on query embeddings. We analyzed the Euclidean distances between query pairs in each image. With 300 queries per image, we computed 300*299/2 pairwise distances. We conducted this analysis on 5,000 images from the VG150 val, comparing scenarios with and without self-attention layers.  By employing a paired t-test and calculating Cohen's d, we ensure that our findings are statistically robust and meaningful.
>
> T-statistic: 54.502
>
> P-value: < 1e-3
>
> Overall mean distance:
>
> - With SA: 7.55
> - Without SA: 7.23 (-5%)
>
> Cohen's d: 0.771
>
> The large T-statistic (54.502) and extremely small p-value (1e-3) indicate that the difference between the two conditions is highly statistically significant. In addition, Cohen's d  = 0.771, approaching the threshold for a large effect (0.8).  These statistical results demonstrate that HydraBranch, by removing self-attention, effectively reduces query distances and encourages duplicated predictions, thereby enhancing our One-to-Many assignment strategy as intended in our design. We add analysis in Sec. 3.4 in the revised manuscript.

---

> ### Author Response · Authors · 2024-11-23
>
> Q3: Does faster convergence come from more diverse samples or more training data within an epoch?
>
> A3: From our perspective, the faster convergence comes from a “better utilization” of training images. **Firstly**, our training process is *exactly the same* as that of previous works (same batch size, total training images), and does NOT generate more training images within an epoch. **Secondly**, compared to other Hungarian matching-based SGG methods [ref6, ref7], our framework generates more positive samples by mining false negative samples. **The total number of training samples remains unchanged**. The performance gains are from better, high-quality training samples in each optimization step. Therefore, the comparison is **fair** and our improvements are credible.
>
> Q4: Adding comparisons with unbiased SGG methods or methods that adopt unbiasing strategies.
>
> A4: We had already included comparisons between Hydra-SGG and methods that adopt debiasing strategies, such as CFA [ref11], in Table 3 of our submission. Debiasing techniques represent an **orthogonal research direction** to our work. They specifically address data imbalance issues and **typically improve mRecall scores**.
>
> As requested, we add additional comparisons with debiasing methods including TDE [ref4],  NICE [ref7], IETrans [ref8], CFA [ref9], and VETO [ref10], NICEST [ref11] in the table below.
>
> | Method | mR@50 | mR@100 |
> | --- | --- | --- |
> | TDE$_{CVPR20}$ | 9.2 | 11.1 |
> | NICE$_{CVPR22}$ | 10.4 | 12.7 |
> | IETrans$_{ECCV22}$ | 12.5 | 15.0 |
> | CFA$_{ICCV23}$ | 12.3 | 14.6 |
> | VETO$_{ICCV23}$ | 10.6 | 13.8 |
> | NICEST$_{TPAMI24}$ | 10.4 | 12.4 |
> | Hydra-SGG | **16.0** | **19.7** |
>
> As seen, Hydra-SGG achieves very competitive performance compared to these specialized debiasing methods. Note that, our method does not incorporate any debiasing strategies, therefore this comparison is **unfair**.  We add analysis in Table 4 in the revised manuscript.
>
> **Ref:**
>
> [ref1] Faster R-CNN: Towards Real-Time Object. NeurIPS 2015.
>
> [ref2] Training Region-based Object Detectors with Online Hard Example Mining. CVPR2016.
>
> [ref3] Focal Loss for Dense Object Detection. ICCV 2017.
>
> [ref4] Unbiased scene graph generation from biased training.
>
> [ref5] SGTR: End-to-End Scene Graph Generation With Transformer. CVPR 2022.
>
> [ref6] RelTR: Relation Transformer for Scene Graph Generation. TPAMI 2023.
>
> [ref7] The Devil is in the Labels: Noisy Label Correction for Robust Scene Graph Generation.  CVPR 2022
>
> [ref8] Fine-Grained Scene Graph Generation with Data Transfer. ECCV 2022.
>
> [ref9] Compositional Feature Augmentation for Unbiased Scene Graph Generation. ICCV 2023.
>
> [ref10] Vision Relation Transformer for Unbiased Scene Graph Generation. ICCV 2023.
>
> [ref11] Noisy label correction and training for robust scene graph generation. TPAMI 2024.

---

> > ### Author Response · Authors · 2024-11-25
> > **Call for open dialogue**
> >
> > Dear reviewer KZeA,
> >
> > Your insightful comments are greatly appreciated. We have provided point-by-point responses to your concerns and are eager to engage in an open dialogue regarding them. Looking forward to hearing from you.
> >
> > Thanks.
> >
> > Authors.

---

> > > ### Comment · Reviewer_KZeA · 2024-11-26
> > >
> > > I still insist the technical solutions appear to be intuitive and empirical. For Q3, I still have concerns that the fast convergence may come from the ratio between positive/negative samples. Have you tried to use the same number of positive samples and negative samples during training to test whether the effectiveness comes from the new generated positive samples? Besides, the performance of the compared unbiased methods is a bit lower than I expected. Therefore, I keep my score.

---

> > > > ### Author Response · Authors · 2024-11-27
> > > >
> > > > Q1: Does faster convergence come from the ratio between positive/negative samples.
> > > >
> > > > A1: We believe there might be some misunderstandings here. First, we clarify how samples are handled in one-stage DETR-based SGG models [ref1, ref2, ref3]. This isn't a conventional two-stage SGG scenario where we can manually adjust the positive/negative sample ratio.
> > > >
> > > > In one-stage DETR-based SGG frameworks, the positive/negative sample ratio is determined by Hungarian matching process, not by manual selection.
> > > >
> > > > The process in a single training step works as follows:
> > > >
> > > > 1. A fixed number of relation queries (e.g., 100) are predefined and fed into the decoder.
> > > > 2. These relation queries interact with image features in the decoder.
> > > > 3. Hungarian matching then assigns ground truth (GT) relation labels to the most suitable relation queries.
> > > > 4. Due to Hungarian matching's one-to-one constraint, if an image has 8 GT relations, exactly 8 relation queries will be matched as positive samples.
> > > > 5. The remaining 92 relation queries are automatically assigned with "no-relation" and become negative samples.
> > > >
> > > > Table 4a (Rows 1 and 2) provides strong empirical evidence for the effectiveness of our approach: while our One-to-One Relation Assignment baseline (Row 1) achieves mR@20/50 of 8.7/12.9 in 50 epochs, Vanilla Hydra-SGG (Row 2, i.e., baseline + Hybrid Relation Assignment) significantly improves the performance to mR@20/50 of 9.9/14.2 in just 12 epochs. **While this does effectively increase the positive/negative ratio**, it is achieved through our matching strategy **rather than manual sample adjustment**. Notably, compared to the baseline, Vanilla Hydra-SGG provides 65.5% (L268, L280) more positive samples on average. The faster convergence comes from allowing more queries to learn from the same GT labels, providing richer supervision signals in each training step.
> > > >
> > > > Q2: Additional comparisons with the latest unbiased SGG methods.
> > > >
> > > > A2: Following the suggestion from reviewer o4Ve, we have expanded our evaluation metrics to include Recall, mRecall, and F-Recall, as they provide "a more comprehensive assessment of Hydra-SGG's effectiveness relative to state-of-the-art approaches.” F@K, which is the harmonic average of R@K and mR@K, serves as an overall metric to enable direct comparisons between different methods that may have varying trade-offs between R@K and mR@K [ref6].
> > > >
> > > > | Method | R@50 | R@100 | mR@50 | mR@100 | F@50 | F@100 |
> > > > | --- | --- | --- | --- | --- | --- | --- |
> > > > | TDE$_{CVPR20}$ | 17.3 | 20.8 | 9.2 | 11.1 | 12.0 | 14.5 |
> > > > | NICE$_{CVPR22}$ | 27.8 | 32.0 | 10.4 | 12.7 | 15.1 | 18.2 |
> > > > | IETrans$_{ECCV22}$ | 25.5 | 29.6 | 12.5 | 15.0 | 16.8 | 19.9 |
> > > > | CFA$_{ICCV23}$ | 27.7 | 32.1 | 12.3 | 14.6 | 17.0 | 20.1 |
> > > > | EICR$_{ICCV23}$ | 26.0 | 30.1 | 15.2 | 17.5 | 19.2 | 22.1 |
> > > > | VETO$_{ICCV23}$ | 28.6 | 34.0 | 10.6 | 13.8 | 15.5 | 19.6 |
> > > > | NICEST$_{TPAMI24}$ | 28.0 | 32.4 | 10.4 | 12.4 | 15.2 | 17.9 |
> > > > | DPL$_{ECCV24}$ | 27.0 | 31.4 | 13.2 | 15.9 | 17.7 | 21.1 |
> > > > | SBG$_{ECCV24}$ | 24.5 | 28.5 | 15.6 | 18.0 | 19.1 | 22.1 |
> > > > | Hydra-SGG | 28.4 | 33.1 | **16.0** | **19.7** | **20.5** | **24.7** |
> > > >
> > > > As seen, Hydra-SGG achieves the highest mR@50, mR@100, F@50, and F@100 among all compared methods. This indicates that our method achieves state-of-the-art overall performance, and again, such comparisons are not fully fair.
> > > >
> > > > **Ref:**
> > > >
> > > > [ref1] RelTR: Relation Transformer for Scene Graph Generation. TPAMI 2023.
> > > >
> > > > [ref2] SGTR: End-to-End Scene Graph Generation With Transformer. CVPR 2022.
> > > >
> > > > [ref3] DSGG: Dense Relation Transformer for an End-to-end Scene Graph Generation. CVPR 2024.
> > > >
> > > > [ref4] Unbiased scene graph generation from biased training. CVPR 2020.
> > > >
> > > > [ref5] The Devil is in the Labels: Noisy Label Correction for Robust Scene Graph Generation.  CVPR 2022.
> > > >
> > > > [ref6] Fine-Grained Scene Graph Generation with Data Transfer. ECCV 2022.
> > > >
> > > > [ref7] Compositional Feature Augmentation for Unbiased Scene Graph Generation. ICCV 2023.
> > > >
> > > > [ref8] Environment-Invariant Curriculum Relation Learning for Fine-Grained Scene Graph Generation. ICCV 2023.
> > > >
> > > > [ref9] Vision Relation Transformer for Unbiased Scene Graph Generation. ICCV 2023.
> > > >
> > > > [ref10] Noisy label correction and training for robust scene graph generation. TPAMI 2024.
> > > >
> > > > [ref11] Fine-Grained Scene Graph Generation via Sample-Level Bias Prediction. ECCV 2024.
> > > >
> > > > [ref12] Semantic Diversity-aware Prototype-based Learning for Unbiased Scene Graph Generation. ECCV 2024.

---

### Official Review · Reviewer_o4Ve · 2024-11-06

**Soundness:** 2
**Presentation:** 3
**Contribution:** 2
**Rating:** 5
**Confidence:** 4

**Summary:**

The manuscript introduces Hydra-SGG, a novel one-stage Scene Graph Generation (SGG) method designed to address the challenges of sparse supervision and false negative samples inherent in DETR-based SGG models. The authors propose a Hybrid Relation Assignment strategy that combines One-to-One and IoU-based One-to-Many Relation Assignments to increase positive training samples and mitigate sparse supervision issues. Additionally, the paper introduces Hydra Branch, an auxiliary decoder without self-attention layers, to enhance the One-to-Many Relation Assignment by promoting duplicate relation predictions among different queries. The model achieves state-of-the-art performance on multiple datasets, including VG150, Open Images V6, and GQA, with a significant reduction in training epochs compared to previous methods.

**Strengths:**

**1.Clear Explanation of Methodology:** The paper does an excellent job of explaining the methodology in a step-by-step fashion. The One-to-One Assignment, One-to-Many Assignment, and HydraBranch are each discussed with sufficient detail, making it easier for readers to follow the technical contributions. Furthermore, the inclusion of pseudocode and diagrams helps clarify how each component interacts within the framework.

**2.Comprehensive Experimental Validation:** The paper provides extensive experiments on two widely-used datasets, Visual Genome, OpenImage, GQA, and demonstrates that Hydra-SGG, consistently outperforms existing baselines. The authors also conduct detailed ablation studies, examining the impact of various components (Threshold T, Training Epoch, Number of Relation Queries). The results substantiate the claim that the framework is a substantial improvement over current model.

**Weaknesses:**

**1.Lack of Novelty:** The motivation and methodology of this work closely resemble those presented in [3], raising questions about its originality and distinct contributions. In particular, [3] introduced the one-to-many relation query matching technique, known as Quality-Aware Multi-Assignment, which enhances training efficiency by allowing a single ground truth (GT) to correspond to multiple sufficiently close predictions or queries. While this work makes a slight modification by removing self-attention among relation queries, this adjustment alone does not constitute a significant advancement in one-stage SGG research. As a result, it is difficult to regard this work as a notably innovative contribution. Further discussion is warranted to clarify how this work differentiates itself from existing approaches.
**2.Perverse Experimental Results:** The rationale for assigning ground truth (GT) to multiple sub-optimal queries is based on label softening and knowledge distillation. However, applying this approach to scene graph datasets with long-tailed distributions (e.g., Visual Genome, OpenImage, PSG) introduces significant challenges. This method tends to overemphasize common relationships (e.g., on, of, has) in the annotations, which can lead to model overfitting on these frequent relationships. Consequently, this emphasis on common relationships undermines the model’s ability to accurately capture rare relationships (e.g., standing on, walking on, lying on), ultimately lowering average recall performance. Given these limitations, a decline in mRecall performance would be expected for such datasets. Surprisingly, however, the experimental results in Table 1 show the opposite. Since this study does not incorporate any specific solutions to address data imbalance, the observed improvement in mRecall is unexpected. This outcome appears to contradict the foundational assumptions of the proposed Hydra-SGG approach, indicating a need for further analysis.

**3.Inadequate Model Performance:** Despite showing a substantially lower R@100 score than the DSGG method [6], the Hydra-SGG model demonstrates a corresponding improvement in the mR@100 metric. Based on prior research [5], incorporating the F-Recall metric would provide a more comprehensive assessment of Hydra-SGG’s effectiveness relative to state-of-the-art approaches. Additionally, the paper lacks a thorough comparison with other relevant studies [2].

**Ref:**

[1] Pair then Relation: Pair-Net for Panoptic Scene Graph Generation. Arxiv 2023

[2] Structured Sparse R-CNN for Direct Scene Graph Generation. CVPR 2022

[3] Groupwise Query Specialization and Quality-Aware Multi-Assignment for Transformer-based Visual Relationship Detection. CVPR 2024

[4] Not All Relations are Equal:Mining Informative Labels for Scene Graph Generation. CVPR 2021

[5] Fine-Grained Scene Graph Generation with Data Transfer. ECCV 2022

[6] DSGG: Dense Relation Transformer for an End-to-end Scene Graph Generation. CVPR 2024

**Questions:**

please refer to the weakness part.

---

> ### Author Response · Authors · 2024-11-23
>
> We thank reviewer o4Ve for the valuable time and thoughtful feedback. We provide a point-to-point response below.
>
> Q1: This work lacks novelty, especially compared to SpeaQ [ref11].
>
> A1: Sorry for this confusion. First of all, we want to clarify that one-to-many assignment is a fundamental concept that has been extensively applied in the community [ref1-ref10] for decades. SpeaQ [ref11] was NOT the first to propose and apply one-to-many assignment. Compared to SpeaQ [ref11], our motivation (L 90) (addressing slow convergence caused by sparse supervision *vs* addressing the problem of unspecialized relation queries), and core techniques (L 285) (mining false negative samples to increase positive samples for each training step *vs* dividing queries and relations into disjoint groups to train specialized queries for specific relations),  implementation details (L 315) (architectural design and training framework *vs* query grouping and assignment strategies) are **DIFFERENT**.
>
> Furthermore, Hydra-SGG differs from other existing one-stage DETR-based SGG models. Existing DETR-based SGG methods focus on designing queries [ref12, ref15], architectures [ref13],  lightweight frameworks [ref14], etc. We emphasize that Hydra-SGG's motivation is to address the slow convergence (L 090) problem in one-stage DETR-based SGG models. Methodologically, Hydra-SGG approaches this through a hybrid query-label assignment and architectural design (L 237 and L285). Hydra-SGG significantly improves the training efficacy of one-stage SGG models, achieving SOTA performance on three common SGG datasets with **only 12 epochs** (VG150 16.0 mR@50, Open Images V6 50.1 weighted score, and GQA 12.7 mR@50). Given such fresh insights and the substantial improvements demonstrated, this study deserves to be shared with our community.
>
> Q2: Removing self-attention among relation queries does not constitute a significant advancement in one-stage SGG research.
>
> A2: Sorry for the misunderstanding. Our main contribution is proposing a comprehensive framework that **addresses the slow convergence** problem in one-stage DETR-based SGG models (L 100). Concretely, we propose a Hybrid Relation Assignment to increase relation supervision signals and HydraBranch, a decoder specifically designed to synergize with our proposed assignment strategy. Removing self-attention in HydraBranch is intentionally designed to encourage different queries to predict duplicated relations (L 288), enhancing our relation assignment strategy. **This design serves specifically our Hybrid Relation Assignment strategy rather than being a general advancement for one-stage SGG research**.

---

> ### Author Response · Authors · 2024-11-23
>
> Q3: Unexpected Improvement in mRecall.
>
> A3: We speculate the improvement in mRecall can be attributed to two main aspects:
>
> 1. Shorter training epochs reduce overfitting:
> Our model trains for only **12 epochs**, compared to SpeaQ's **52 epochs**. This shorter training period mitigates the risk of overfitting to high-frequency (common) relations. Furthermore, the relatively higher mRecall and lower overall Recall demonstrate that our model does not overfit to frequent ones.
>
>   2.  Our One-to-Many assignment enhances training signals for rare relations:
>
> While One-to-Many Assignment increases the number of queries for both rare and common relations, it **avoids overemphasizing common relationships**. As mentioned in line 259, we assign a fixed **6 queries per relation category**. For example, in an image with 10 "on" relations and 2 "sit" relations:
>
> - **One-to-One Assignment** would allocate **10 queries to "on"** and only **2 queries to "sit"**.
> - **Our One-to-Many Assignment** assigns **6 queries for both "on"** (a **60% increase**) and **6 for "sit"** (a **300% increase**).
>
> Therefore, One-to-Many assignment enhances training signals for rare relations by providing them with a larger relative gain, while avoiding overemphasis on common relations.
>
> Q4: Inadequate model performance and missing F-Recall results.
>
> A4: As requested, we have added F-Recall results and comparisons with SSR-CNN [ref16]. Hydra-SGG achieves SOTA F-Recall with substantially fewer parameters (**67.6**M).  This suggests that Hydra-SGG achieved the highest overall performance. We add comparisons in Table 1 in the revised manuscript.
>
>
>
> | Method | Param (M) | R@50 | R@100 | mR@50 | mR@100 | F@50 | F@100 |
> | --- | --- | --- | --- | --- | --- | --- | --- |
> | SGTR$_{CVPR22}$ | 117.1 | 25.1 | 26.6 | 12.0 | 14.6 | 16.2 | 18.9 |
> | SSR-CNN$_{CVPR22}$ | 274.3 | 32.7 | 36.9 |  8.4 | 10.0 | 13.4 | 15.7 |
> | ISG$_{NeurIPS22}$ | 93.5 | 29.5  | 32.1 | 7.4 |  8.4 | 11.8 | 13.3 |
> | RelTR$_{TPAMI23}$ | 63.7 | 27.5 | 30.7 |  10.8 | 12.3 | 15.5 | 17.6 |
> | DSGG$_{CVPR24}$ | - | 32.9 | 38.5 |  13.0 |  17.3 | 18.6 | 23.9 |
> | SpeaQ $_{CVPR24}$ | - | 32.9 | 36.0 |  11.8 |  14.1 | 17.4 | 20.3 |
> | EGTR$_{CVPR24}$ | 42.5 | 30.2 | 34.3 | 7.9 | 10.1 | 12.5 | 15.6 |
> | Hydra-SGG | 67.6 | 28.4 | 33.1 | **16.0** | **19.7** | **20.5** | **24.7** |
>
>
>
>
> **Ref:**
>
> [ref1] Rich feature hierarchies for accurate object detection and semantic segmentation. CVPR 2014.
>
> [ref2] Fast R-CNN. ICCV2015.
>
> [ref3] Faster R-CNN: Towards Real-Time Object Detection with Region Proposal Networks. NeurIPS2015.
>
> [ref4] You Only Look Once: Unified, Real-Time Object Detection. CVPR2017.
>
> [ref5] Mask R-CNN. ICCV2017.
>
> [ref6] Focal Loss for Dense Object Detection. ICCV2017.
>
> [ref7] OTA: Optimal Transport Assignment for Object Detection. CVPR2021.
>
> [ref8] YOLOX: Exceeding YOLO Series in 2021. arxiv 2021.
>
> [ref9] Group DETR: Fast DETR Training with Group-Wise One-to-Many Assignment. ICCV2023.
>
> [ref10] YOLOv10: Real-Time End-to-End Object Detection. arxiv 2024.
>
> [ref11] Groupwise Query Specialization and Quality-Aware Multi-Assignment for Transformer-based Visual Relationship Detection. CVPR 2024.
>
> [ref12] SGTR: End-to-End Scene Graph Generation With Transformer. CVPR 2022.
>
> [ref13] RelTR: Relation Transformer for Scene Graph Generation. TPAMI 2023.
>
> [ref14] EGTR: Extracting Graph from Transformer for Scene Graph Generation. CVPR 2024.
>
> [ref15] DSGG: Dense Relation Transformer for an End-to-end Scene Graph Generation. CVPR 2024.
>
> [ref16] Structured Sparse R-CNN for Direct Scene Graph Generation. CVPR 2022.
>
> [ref17] Fine-Grained Scene Graph Generation with Data Transfer. ECCV 2022.

---

> ### Author Response · Authors · 2024-11-25
> **Call for open dialogue**
>
> Dear reviewer o4Ve,
>
> First, we want to thank you again for reviewing our work. We have provided point-to-point responses to your comments. However, we have not received any feedback yet since the open discussion phase. We appreciate your new input based on our responses.
>
> Thank you!
>
> Authors.

---

> > ### Author Response · Authors · 2024-11-29
> >
> > Dear reviewer o4Ve,
> >
> > Thank you for taking the time to review our work. We have carefully addressed all your comments and incorporated the suggested revisions. Please let us know if you have any additional feedback or concerns.
> >
> > Thank you!
> >
> > Authors.

---

> ### Author Response · Authors · 2024-12-02
>
> Dear Reviewer o4Ve,
>
> This is a gentle reminder that today (December 2nd, Anywhere on Earth) marks the deadline for review feedback. We are available to address any questions or concerns until tomorrow.
> We have addressed your comments in detail, added new evaluations (F-Recall), and more comparisons, and made significant updates. Please don't hesitate to reach out if you need any clarification.
>
> Best regards,
>
> Authors

---

### Author Response · Authors · 2024-11-23

## Summary of Revisions

To all reviewers:
We sincerely appreciate your time and effort in reviewing our manuscript. Below, we outline the major changes made in response to your feedback:

- We clarify the novelty of Hydra-SGG compared to SpeaQ and other existing one-stage DETR-based SGG models, addressing Reviewer o4Ve's concerns.
- We offer more detailed discussions regarding why mRecall performance is high, according to Reviewer o4Ve's comments.
- We provide comprehensive F-Recall results and comparisons with state-of-the-art methods, including SSR-CNN, per Reviewer o4Ve's request.
- We clarify the analyses conducted to propose Hydra-SGG, addressing Reviewer KZeA's concerns.
- We add detailed statistical analysis on query embeddings to demonstrate how HydraBranch enhances One-to-Many Assignment, responding to Reviewer KZeA's feedback.
- We include additional comparisons with unbiased SGG methods (TDE, NICE, VETO, etc.) showing Hydra-SGG's competitive performance, as requested by Reviewer KZeA.
- We conduct an ablation study on different loss weights between One-to-One and One-to-Many assignments, addressing Reviewer ARZw's question.
- We expand the discussion on limitations and future work, including potential improvements to One-to-Many Relation Assignment strategies, per Reviewer ARZw's suggestion.
- We add additional analyses, ablation studies, and comparisons of rebuttal in the revised manuscript.

Sincerely yours,

Authors.

---

### Meta-Review · Area_Chair_rx9S · 2024-12-20

**Metareview:**

This paper introduces Hydra-SGG, a novel method for scene graph generation that addresses sparse supervision and false negatives by employing a hybrid relation assignment strategy and a Hydra branch decoder. These innovations contribute to improved training and competitive results across several datasets, achieved with relatively few training epochs.

The methodology is acknowledged for its clarity and detail. The effectiveness and efficiency of the proposed approach in tackling critical issues in one-stage scene graph generation, such as sparse supervision and false negatives, are also well-appreciated. The experimental results are robust. However, some concerns and questions were raised during the initial round of feedback, including queries about the originality of the approach compared to existing work, unexpected results in the mRecall metric, insufficient performance metrics, and a lack of thorough analysis and exploration.

The authors have provided detailed clarifications, analyses, and additional experimental results focusing on the F-Recall metric and comparisons with additional baselines. After careful review of the paper, its revisions, and the discussion, I believe the key concerns including model performance and unexpected performance explanation have been adequately addressed. Therefore, I recommend the acceptance of this paper.

**Additional Comments On Reviewer Discussion:**

The original concerns are: 1. lack of novelty, particularly in comparison to SpeaQ, 2. unexpected results for rare relations, 3. inadequate performance metrics, and 4. the lack of theoretical backing, along with insufficient analysis and exploration.

Regarding novelty, the authors highlight that the one-to-many assignment is a fundamental concept not proposed by SpeaQ. More importantly, the motivation, core techniques, and implementation details differ significantly from SpeaQ. In response to the unexpected improvement in mRecall, the authors provide a detailed breakdown using an example and clearly explain that the proposed method avoids overemphasizing common relationships. Regarding inadequate model performance, the authors present additional results on the F-Recall metric and a comparison with SSR-CNN. For theoretical backing, the authors argue that this is not a theoretical paper, but the proposed methodology is systematic, which seems reasonable to me. Given all these points, I believe the majority of the major concerns have been addressed.

---

### Decision · Program_Chairs · 2025-01-22

Accept (Poster)